# Vitamin D and Systems Biology

**DOI:** 10.3390/nu14245197

**Published:** 2022-12-07

**Authors:** Shahid Hussain, Clayton Yates, Moray J. Campbell

**Affiliations:** 1Division of Pharmaceutics and Pharmaceutical Chemistry, College of Pharmacy, The Ohio State University, Columbus, OH 43210, USA; 2Department of Biology and Center for Cancer Research, Tuskegee University, Tuskegee, AL 36088, USA; 3Department of Pathology, Johns Hopkins University School of Medicine, Baltimore, MD 21287, USA; 4Department of Oncology Sidney Kimmel Comprehensive Cancer Center, Johns Hopkins University School of Medicine, Baltimore, MD 21287, USA

**Keywords:** vitamin D receptor, systems biology, cell differentiation, prostate cancer

## Abstract

The biological actions of the vitamin D receptor (VDR) have been investigated intensively for over 100 years and has led to the identification of significant insights into the repertoire of its biological actions. These were initially established to be centered on the regulation of calcium transport in the colon and deposition in bone. Beyond these well-known calcemic roles, other roles have emerged in the regulation of cell differentiation processes and have an impact on metabolism. The purpose of the current review is to consider where applying systems biology (SB) approaches may begin to generate a more precise understanding of where the VDR is, and is not, biologically impactful. Two SB approaches have been developed and begun to reveal insight into VDR biological functions. In a top-down SB approach genome-wide scale data are statistically analyzed, and from which a role for the VDR emerges in terms of being a hub in a biological network. Such approaches have confirmed significant roles, for example, in myeloid differentiation and the control of inflammation and innate immunity. In a bottom-up SB approach, current biological understanding is built into a kinetic model which is then applied to existing biological data to explain the function and identify unknown behavior. To date, this has not been applied to the VDR, but has to the related ERα and identified previously unknown mechanisms of control. One arena where applying top-down and bottom-up SB approaches may be informative is in the setting of prostate cancer health disparities.

## 1. Systems Biology and Biomedicine

### 1.1. The Opportunities of Applying Systems Biology Approaches in Biomedical Research

Much of the discovery in biomedicine has been centered on the classic paradigms of reductionist biology, in which phenotypes are interpreted as the interaction of either single or small groups of molecules. Across biomedical sciences, there are many remarkable examples of how this approach has led to the discovery of drivers of human disease, and equally remarkable examples of new therapies designed to target these drivers. For example, high-profile examples of this reductionist approach include the discovery of the oncogenic role of the BCR-ABL fusion gene in the etiology of chronic myeloid leukemia [1] and the development of a targeting kinase inhibitor such as Imatinib [2].

This is a striking example of the so-called bench-to-bedside research and represents one of the earliest examples of precision medicine. At times, however, this reductionist approach can appear limited both theoretically and clinically as the etiology of Imatinib-resistant phenotypes only too well demonstrates [3]. Across cancers and other disease phenotypes, a stumbling block can be identifying single strong disease driver mechanisms, which in turn accurately predict drug sensitivities and therapeutic effectiveness. Therefore, delivering a fuller prediction of disease drivers and therapeutic vulnerabilities may require developing different methodologies. Ideally, any methodologies would also take advantage of the ever-increasing stream of high-dimensional biological data to inform diagnosis and prognosis. Perhaps these approaches (SB and reductionist) can actually be highly symbiotic.

Systems biology (SB) aims to apply mathematical approaches to build a predicative and quantitative model of biological systems, with the goal to use model predictions to define specific physiological or pathophysiological states and outcomes. The models derived from such SB approaches applied to experimentally derived biological data aim to identify the dynamic behavior of networks that are the center of cellular behaviors [4,5]. These approaches are readily scalable and not restricted by scope. SB approaches can be applied to discrete cell signaling systems, such as gene regulatory networks and signal transduction cascades, to cell–cell interactions, tissue organization, organismal behavior, and to complex multi-organism interactions as seen within, for example, the function of the gut and even complete ecosystems [6].

Furthermore, the models built by SB approaches aim to define the functioning of living organisms not solely by looking at the constituent molecules such as DNA, RNA, proteins, and metabolites but rather by the process-level biological systems they constitute, such as mitosis and metabolic control [7,8]. A key hallmark of these models is that they capture the states of the system in a predicative and quantitative manner and can be exploited to drive novel understanding. As these in silico models can be interrogated rapidly, multiple components of any given model can be dissected to reveal unintuitive findings [9,10,11].

A consequence of the integrated nature of biological signaling is the emergent complexity, which underpins human health. For example, the dexterous control of transcriptional networks is derived from a high ratio of transcription factors to regulated mRNA or miRNA targets. This ratio, combined with large regulatory regions, results in unparalleled plasticity over the choice, amplitude, and period of transcription [12,13]. More specifically, transcription factor modules recognize, interpret and sustain histone modifications and ultimately establish boundaries between transcriptionally rich euchromatin and transcriptionally restricted heterochromatin. These boundaries are cemented further by the regulation of CpG island methylation; these two processes are dynamically intertwined. Again, from an SB perspective, it is reasoned that modeling these transcriptional processes will help to explain the highly integrated nature and robustness of normal transcriptional control, whereby the processes have redundancies such that they do not radically alter in response to external signals and do not fail when a single component is altered. By contrast, transcriptional networks in cancer cells display a loss of transcriptional plasticity and do not display the full breadth of signaling capacities. The evolution of the malignant transcriptome is seen clearly in the nuclear receptor and MYC superfamilies (reviewed in [14]).

The concept of SB builds on so-called “holistic biology” developed in the 1960s and coupled with informatics theory and modeling approaches developed through the 20th century. However, the application of SB methodologies and the expansion of SB concepts in biomedicine has been boosted since the early 2000s by the technological advances in the development of high dimensional data approaches frequently derived from next-generation sequencing technologies coupled with bioinformatic approaches. This progression from the 1950s to the current state has been profoundly catalyzed by the human genome project, with its draft sequence published in 2001, and a final reference genome published in 2022 [15] alongside other reference genomes across the animal and plant kingdom. In this manner, the combination of high throughput experimental approaches in the wet lab coupled with complex statistical analyses and computational methods in the dry lab have led to a more comprehensive characterization of multiple diverse organisms, and a shift of focus from molecules to their interactions and the networks they form [16]. This sophistication and power of prediction are most likely set to increase when SB models also include a spatio-temporal characterization of cell behavior, including the dynamics of how molecules are exchanged between compartments and exported from the cell. Again, modeling these aspects of cell behavior has been massively impacted by an explosion in single-cell and spatial technologies.

There are several well-justified advantages of applying SB approaches, which have the genuine potential to complement and extend the reductionist paradigm. Expressing the complete interactions within biological systems in mathematical terms reduces the impact of biases introduced by focusing on single proteins, and instead can reveal under-explored control points in the system. Furthermore, expressing biological systems in the context of processes, rather than well-understood individual components, has the potential to assimilate more readily new high dimensional data generated in biological experiments. Consequently, it is reasonably anticipated that significant, novel, and unpredicted strides will be made in understanding the control and responsiveness of biological events, which in turn can generate new insights into disease susceptibility and therapeutic opportunities [17,18].

### 1.2. Systems Biology Builds upon an Asymptotic Recursion between Wet Lab and Dry Lab

At its core, SB approaches have an asymptotic recursion between research activities in the wet lab and dry lab. Models are built in the dry lab that reflects the biological observations in the wet lab and are used to generate predictions for how the system can behave. Such predictions are then formulated into testable interventions in the wet lab to generate data for model refinement in the dry lab. It is this oscillation between wet and dry labs that is so potentially powerful, but at the same time so daunting. Broadly, two strategies have emerged to develop such models, by either top-down or bottom-up approaches (Figure 1).

In the top-down approach, large datasets are interrogated with statistical methods to find patterns in the data with which to derive predictions on the system organization [19]. The models in top-down SB are phenomenological, meaning they are not directly mechanistically based and do not require knowledge about relationships between different molecular components. Identification of significant associations is used to develop a hypothesis on the nature of the molecule interactions and associations identified. These approaches naturally lend themselves to omics-derived data and develop hypotheses to be tested by wet lab analysis [20]. It is worth noting that the identified associations that are significant may not be causal, and in fact, themselves may not be true. Top-down approaches are often used with subsystems that have not yet been characterized to a high level of mechanistic detail and approaches such as Bayesian modeling are appropriate due to the missingness in the data from biological regulatory networks [21].

By contrast, the bottom-up approach begins with the hypothesis of biological mechanism and formulating equations on how the components of the system interact and then running simulations to generate predictions. This approach relies on experimental studies to determine the kinetic and chemical properties of the components, and starts with formulating system behaviors in rate equations, for example, expressed as differential equations, of the constituting parts of each system. These formulations are then integrated to predict the system behavior, with the goal to combine pathway models into a model for a larger system [21]. The data on the system under study are subjected to perturbations in the cell context and models are refined from the data. An example of a bottom-up approach is the silicon cell program where computational replicas of actual pathways are made to calculate system behavior [22].

The power of these approaches arises from being able to identify and direct experiments to test fundamental questions of a biological system. From the top-down perspective, these questions include identifying in a genome-wide manner the interactions of all components in a system to define the metabolic control that ultimately brings about cell, tissue, or organism behavior. Arising from this approach questions can be asked of a system. For example, what is the interconnectedness of the system and how does that change between health and disease states, or in different development or differentiation states? Which hubs in such networks are central and which are peripheral? Again, how does hub distribution shift in disease and development transitions? How do changes in gene and protein expression combine to control metabolism? How do germline or somatic structural variants change network topology and metabolic flux?

Similarly, from the bottom-up perspective, a model is curated from known biological interactions and expressed in mathematical terms to test behavior and make new predictions. For example, with any signaling system how is activation controlled and silenced? Given the ubiquitous role of enzymes in biology, a common question is how does changing the kinetics of activating/de-activating enzymes lead to altered signal amplitude? To what extent do germline or somatic structural variants in enzymes, or co-factors modulate enzymatic capacity and what effect does that exert on signal strength? More broadly, how are different signaling systems integrated and what is the quantitative effect of convergence? Does crosstalk generate synergistic events in regulatory complexes or are they merely convergent downstream at endpoints? What is the magnitude of control at each step and is there an order to control ranging from find-tuning to signal-independent activation? Finally, questions can be asked of the system in disease or development states and tested for pharmacological relevance including identifying which targets may be impactful but have deleterious side effects.

## 2. The Opportunities and Challenges of Applying Systems Biology Approaches to Studying the Vitamin D Receptor

A systems-level appreciation for vitamin D signaling has existed for several centuries, given that a description of rickets was first described in the 17th century [23]; rickets arises from impaired bone mineralization due to insufficient signaling via the vitamin D receptor (NR1I1/VDR) (reviewed in [24]).

The VDR is a Type II member of the nuclear receptor superfamily, which binds the active hormone 1α,25(OH)_2_D_3_. Several features of how the VDR functions are important from an SB perspective. A feature of Type II receptors such as the VDR is that independent of ligand the receptor is significantly associated with the genome bound to cis-regulatory elements (CRE) and may exert repressive effects, which is reversed by ligand activation leading to genomic redistribution and transactivation [25,26,27,28]. Additionally, of interest from a systems perspective is that VDR interacts in distinct ways with a number of proteins. Firstly, it heterodimerizes with the RXRs (NR2B1/RXRα, NR2B2/RXRβ, NR2B3/RXRγ), which is also a central dimer partner for other Type II receptors including those retinoic acid receptors (NR1B1/RARα, NR1B2/RARβ, NR1B3/RARγ), and the peroxisome proliferator-activated receptors (NR1C1/PPARα, NR1C2/PPARβ, NR1C3/PPARγ) [reviewed in [24]]. Secondly, the VDR interacts with a range of coregulator proteins such as coactivators and corepressors that exert antagonist roles in the control of local chromatin structure, as well as components of the SWI/SNF complexes to remodel nucleosome positioning and other enzymes such as helicases [29,30] and splicing factors [31] to also facilitate transcription. The VDR appears to participate in protein–protein–DNA interactions, for example with other transcription factors, in a trans-regulatory mechanism to regulate transcription. Finally, there is emerging evidence for the VDR, like other nuclear receptors [32,33], to interact functionally with long non-coding RNAs (lncRNA) to control gene expression by directly affecting the DNA environment or other RNA binding proteins [34]. Indeed one such lncRNA, steroid receptor activator (SRA) coimmunoprecipitates with the coactivator NCOA1/SRC1 and may function more broadly as a scaffold for NR complexes (reviewed in [32]).

Finally, of central importance to the complete VDR signaling system is the generation of the ligand, 1α,25(OH)_2_D_3_, the levels of which are dynamically controlled in terms of synthesis of the precursor 25(OH)D_3_ which is held in a relatively tight range in the serum, and signs of deficiency in the VDR system can be seen when this serum level is diminished. The various metabolic steps that lead to the generation of 25(OH)D_3_, the active ligand 1α,25(OH)_2_D_3_, and a range of metabolites that ultimately lead to its catabolism are all tightly controlled by enzymatic reactions both in an endocrine and tissue-specific local intracrine manner.

### 2.1. Top-Down Approaches Applied to VDR Biology

From a top-down SB perspective, it is interesting to ask in what context the VDR itself, or coregulators, or ligand generation are identified as a central hub in gene networks associated with different cell phenotypes and disease states, and from a bottom-up SB perspective, it is attractive to develop models that accurately capture the cycling interactions of the VDR with different co-factor and predict how this relates to diverse transcriptional outputs, again across cell phenotypes and disease states.

High dimensional data approaches have been applied to ask questions about what the transcriptional effects are of adding 1α,25(OH)_2_D_3_, or deleting the VDR. From a top-down SB perspective, the question of VDR function is alternatively phrased to identify biological circumstances where the VDR is identified as a hub in the transcriptional network independent of any knowledge of the system. In this manner, analyses of across myeloid cells [35], granulocytes [36], and megakaryocytes [37] have identified significant control functions for the VDR to regulate specific cell differentiation outcomes. Interestingly, these studies support some of the earliest studies that explored the functions of 1α,25(OH)_2_D_3_ and revealed its capacity to initiate differentiation of leukemia cells [38,39]. These findings might suggest a role for the VDR to be mutated or deleted in leukemia and although there were numerous candidate studies of the VDR expression and genomic integrity, large-scale genomic approaches, for example in the TCGA leukemia cohort [40], suggest that the VDR is neither distorted to act as cancer-driver, nor has it been therapeutically exploited to date.

Together, these findings suggest that whilst the VDR is biologically impactful in the normal differentiation, for example, the process to monocytes, it is not itself so frequently disrupted in leukemia such that when it is mutated it acts in an oncogenic manner to disrupt normal progenitor differentiation, and it cannot be targeted to induce leukemia cell differentiation in patients. At first glance, these findings may appear contradictory, but most likely reflect the nature of redundancy in the system, and the role of VDR to undergo diverse protein interactions. Specifically, the VDR is known to interact with CEBPs [41] to induce leukemia cell differentiation [41], and indeed the CEBPs are a master regulator in the physiological transcriptional module containing the VDR to regulate monocyte differentiation [35]. However, CEPPs interact with multiple different nuclear receptors to regulate cell fates [42,43,44,45], and redundancy in the system limits the impact that altered VDR function alone can exert on cell fates. Finally, it is interesting to note that CEBPA mutations are in the top 33 mutational events in the TCGA leukemia cohort [40].

These identified roles for the VDR in monocyte differentiation reflect a broader function for the receptor to signal in differentiation processes [46] and immune phenotypes. Alongside the roles of vitamin D to prevent rickets, in the late 19th century the concept emerged of sunlight therapy, known as heliotherapy, to fight tuberculosis and other infections [47]. Indeed, unbiased transcriptomic analyses of the impact of *Mycobacterium tuberculosis* demonstrated a role for Toll-like receptors to initiate a signal transduction cascade in macrophages that upregulated the VDR and induced an antimicrobial response. Intriguingly, this response appeared to be dampened in people with African genomic ancestry perhaps associated with low serum levels of 25(OH)D_3_ [48]. Similarly, *Mycobacterium tuberculosis* infection of lung cells identified VDR as a key regulated transcription factor [49]. Another infection of interest is the coronaviruses and in 2013 the lung cell response towards severe acute respiratory syndrome (SARS) arising from a coronavirus infection again identified the VDR as a key regulator of response [50]. More recently, COVID-19 infection of T cells from bronchoalveolar lavage demonstrated a role for the VDR to trigger super-enhancer activation and the control of innate immunity [51]. An unbiased identification of the role of the VDR in the regulation of innate immunity and immune phenotypes has also been identified by GWAS [52,53,54,55].

Viewed from the perspective of unbiased top-down statistical analyses has revealed critical roles for the VDR in innate immunity, control of inflammation, regulation of differentiation, and metabolic control [56,57,58], but not in some of the other phenotypes that are investigated at the pre-clinical and candidate level, notably including cancer; this was established in a previous structured literature search (reviewed in [59]). That is, no cancer-associated GWAS identifies structural variants in the VDR associated with cancer risk, and none of the TCGA papers (over 10,000 tumor samples across more than 30 tumor types) identify frequent VDR-associated structural variants or significantly altered expression that associates with cancer clinical phenotypes.

GWAS studies have continued to evolve in terms of the complexity of study design [60] and cohort sizes and different genomic ancestry, and methods have also developed to finding interactions between variants and also considering how non-coding variants withing regulatory regions may determine how efficiently a given transcription factor can function; so-called a cis-expression quantitative trait loci (eQTL). Certainly, there is evidence for significant enrichment of germline structural variants in VDR binding sites associated with genes that regulate immune phenotypes [61]. Likewise, there is evidence for such eQTLs are significantly impacted by genomic ancestry, for example in the case of the ex vivo response towards 1α,25(OH)_2_D_3_ in normal colon cells [62].

These methodological developments and the ever-increasing repertoire of high dimensional data are likely to increase the understanding of VDR and vitamin D signaling phenotypes and will most likely become ever more important to develop new hypotheses over how signaling occurs and where it is most biologically relevant. Perhaps this is readily illustrated by the application of Mendelian randomization methodology to test the factors such as levels of serum 25(OH)D_3_ (either measured or predicted) and their causal relationship with different clinical phenotypes. In this manner 25(OH)D_3_ levels and components of the VDR axis are significantly casual in multiple sclerosis, hand grip strength, bacterial infection, and type 2 diabetes [63,64,65,66], but not for the prevention of COVID-19 infection, birth weight, or colon cancer [58,67,68].

### 2.2. Bottom-Up Approaches Applied to VDR Biology

Applying a bottom-up SB approach to VDR-dependent gene regulation requires constructing a kinetic model that captures the function of different VDR-containing complexes and the quantitative regulation patterns of VDR target genes. Such a model would combine spatial-temporal measurements for the generation of 1α,25(OH)_2_D_3_ in a target tissue, VDR genomic interactions, such as initial concentrations in the nucleus of the proteins involved, their half-life, and their associations, to explain how these actions predict the downstream transcriptional impact. Model refinement and validation can then be undertaken by empirical measurements of the cycling of VDR genomic interactions and transcriptional outputs.

The working assumption is that the transcription factor complex contains enzymes that remodel chromatin to allow signaling to the RNA polymerase complex and the initiation of transcription. In the case of a transcription factor such as the VDR, which has nuclear residence independent of ligand exposure, the complex exists in two states of genomic interaction, namely with and without ligand, and in each with qualitatively and quantitatively different cofactors, and also in terms of distribution (distal or proximal). The output of the model will then be to predict the kinetics of mRNA accumulation in a manner that reflects the different ligand-dependent and independent states, and spatial association to target genes. Of course, these components are most likely too simple to predict mRNA accumulation accurately. However, even such a simple model for different genes and the VDR could begin to define how much of the difference between mRNA accumulation could be explained by merely considering spatial and temporal factors for transcription factor residence on a gene. Model components could then be added to consider other well-understood biological components such as disease states, and chromatin states, as well as the impact of DNA helicases required for transcription, or even the role of non-coding RNA to impact mRNA accumulation.

To date, such an approach has been applied to several transcription factors in human cells. Within the nuclear receptor superfamily, kinetic modeling of ERα predicted a role of receptor phosphorylation to act as an underexplored feedback mechanism to predict RNA accumulation and was validated by empirical measurements [69]. Outside of nuclear receptors, other human transcription factor-centered models include describing the separate actions of NF-κB [70,71] and p53 [72], and their collective oscillatory behavior [73]. Separately, signal transduction events have also been modeled quite extensively, including the transduction events for RAS [74,75] and its interaction with other regulatory events such as the circadian clock [76] and EGFR signaling [77].

These approaches are powerful, and it is perhaps surprising why they have not been more widely applied to other nuclear receptors beyond the ERα, for example to the clinically relevant VDR. Several impediments no doubt include the time, resources, and interdisciplinary expertise required for model development and refinement. One impediment is establishing collaborations that bring insight from the wet and dry labs. To be successful these collaborations require sufficient data density of dynamic cellular events, such as precise measurements of the concentration of key proteins and RNA molecules in a cell, the frequency of transcription factor genomic interactions, and levels of mRNA accumulation to justify modelling in mathematical terms. Furthermore, these challenges are sometimes impeded by an uneasy relationship between the biological and mathematical communities [5]. The incentive for these intensive modelling efforts, most likely involving interdisciplinary collaborations, is that once such a model is built it has the potential to be translated to other transcription factor genome interactions, or across cell types, or disease states and allow speculation on the biological processes without having to undertake time-intensive and costly experiments in the wet lab.

## 3. Prostate Cancer and Health Disparities; An Exemplar of the Opportunities Arising from Systems Biology Approaches in Biomedicine

Men of African genomic ancestry in American (African American, (AA)) experience higher risks of developing more aggressive prostate cancer (PCa) than European American (EA) counterparts [78,79,80,81], which reflects underlying genetic [82,83,84,85] and epigenetic [86,87,88,89,90,91] drivers and biopsychosocial processes [89,92,93]. The role of the glucocorticoid receptor (GR) as a primary target for stress response has more recently been examined in the context of cancer (reviewed in [94]) and offers a potential functional explanation for how stress can be an accelerant of PCa in AA men [94,95,96,97]. A role for VDR signaling appears to be more high profile in the etiology of AA rather than EA PCa. Given that UVB radiation degrades folic acid as well as catalyzing vitamin D synthesis, a strong inverse correlation between skin pigmentation and latitude has arisen during ancestral adaptation [98,99], and amongst AA PCa patients there are significant associations between low serum vitamin D_3_ levels and incidence and progression risks of PCa [100,101,102,103,104,105,106,107,108,109,110,111,112,113,114,115]. Furthermore, although vitamin D_3_ supplementation in the VITAL cohort [116,117] had no overall impact on cancer incidence, the AA participants experienced a suggestive 23% (*p* = 0.07) reduction in cancer risk, indicating that larger cohorts may be more informative [118,119,120]. Strikingly, vitamin D_3_ supplementation in AA and EA PCa patients only significantly modulated prostate gene expression in the AA patients [121], associated with the control of inflammation. This was supported by our recent study [122] in AA and EA cell models which have identified qualitatively and quantitatively distinct VDR actions in terms of VDR protein–protein interactions and more frequent VDR genomic interactions associated with significantly distinct target gene expression.

These data support the concept that one cell function that appears to be most highly responsive in a manner that reflects genomic ancestry is the role of VDR to modulate innate immunity [48,123,124,125,126], and reflecting this there are correlative findings that support a relationship between 25(OH)D_3_ levels and immune-modulatory factors such as interleukin (IL)-6 [127,128]. IL6 release into the serum is associated with activated macrophages and is elevated in Ghanaian and AA men [129,130,131]. Together these studies suggest that the biology of the prostate is the most sensitive VDR signaling in AA men, and more impacted by inadequate VDR signaling either as a result of molecular mechanisms or environments of low 25(OH)D_3_. One biologically impactful consequence of this is the loss of control of inflammatory signals. These studies also highlight that African ancestry is itself divergent, and is also further modified by admixture, for example in the AA population (refs).

### The Potential Application of SB Approaches to Prostate Health and Disease

There are significant knowledge gaps in identifying how genomic ancestry, the environment, and lifestyle choices may combine to impact prostate health and disease. The application of SB modeling could be used to generate biologically plausible hypotheses to test specific relationships in these complex interactions. These knowledge gaps include whether germline structural variants impact the generation of 1α,25(OH)_2_D_3_, and the extent of relationships between serum 25(OH)D_3_ levels and PCa in a manner that reflects genomic ancestry, and if, and to what extent, this is impacted by admixture. In experimental models, it is emerging that the VDR genomic interactions differ between EA and AA prostate cell models (ref), and recently AR genomic interactions also appear different in EA and AA PCa samples. However, it is unclear if either the underlying sequence, for example at enhancer binding sites, or other mechanisms are impacting why these different genomic interactions arise. Finally, given the multi-parameter process of cis-regulatory interactions that drive transcription through to processed RNA and translation to protein, it is also unclear how genomic ancestry impacts these processes potentially in an emergent manner.

Together, these knowledge gaps can be combined in a hypothesis; namely that PCa risks in AA men reflect the interplay of genomic ancestry, including admixture, coupled with altered environmental signals that combine to drive qualitatively and quantitatively distinct functioning of the VDR and potentially other transcription factors and underscore the health disparities in PCa. Potentially, this is also an attractive setting to develop a series of SB models with which to test these interactions in silico and generate predictions to validate in vitro and in vivo (Figure 1).

This provides an opportunity for both top-down and bottom-up SB modeling. From the top-down perspective, high dimensional data are available to test develop models that test how genomic ancestry significantly determines the associations between serum 25(OH)D_3_ levels [118,119,120] and PCa structural variants [132,133], epigenetic states [86,87,88,89,90,91] and gene expression [87,91,134] and how this significantly relates to serum inflammatory markers. More specifically, Mendelian randomization approaches would be appropriate to test how germline structural variants are associated with serum 25(OH)D_3_ levels depending on genomic ancestry.

To complement these approaches, other top-down strategies applied to high dimensional data sets could then determine epigenomes such as CpG methylation, histone modification, and chromatin accessibility [135,136] and transcriptome relationships in AA and EA PCa patients. Construction of a spatial matrix from epigenomic data, co-incidence with prostate cancer-specific enhancers [137,138], will be binned by accounting for orientation and distance to gene features. Machine learning approaches can then be applied to identify and test the significance of relationships between this matrix and PCa transcriptomes. For example, bootstrapping approaches can test the associations of the epigenome and patterns of observed gene expression compared to simulated data by random sampling [139]. In parallel, other approaches such as the Pareto optimization algorithm [140] can define the ordered correlation of relationships between the cistrome matrix and genesets, of which the strongest (positive or negative) are of greatest biological significance. Comparing the results of both approaches within and across genomic ancestry will reveal more significant cistrome–transcriptome features that are potentially driving PCa health disparities. Lasso and ridge regression can then be performed on the most significant cistrome–transcriptome relationship genes to identify gene expression patterns that predict clinical outcomes in publicly available data [141,142].

To meet these top-down approaches, bottom-up methods can be applied to build kinetic models that take into consideration the amount of the VDR and its key interacting factors in AA and EA prostate cells to simulate its ligand-dependent and independent genomic states and distribution across epigenetic states. The goal of this modeling would be to provide biological plausibility for why the VDR appears to be significantly transcriptionally divergent between AA and EA PCa. Such predictions would form the justification for validation in the wet lab using AA prostate cell line models and patient-derived xenografts. Finally, such modeling may well justify tailoring recommendations for healthy vitamin D serum levels that reflect genomic ancestry.

## 4. Summary

The VDR signaling system is physiologically impactful and when its functions are altered it is associated with several disease states. Systems biology approaches to the analyses of VDR functions have been applied in several top-down studies that have identified significant VDR functions. To date, however, no bottom-up approaches to kinetic modeling have been applied. The incentive in biomedicine to continue to apply SB approaches would be to develop insight into what are the most prominent contexts where the VDR system exerts biological impact and in turn how this can be exploited in diagnostic, prognostic, or therapeutic contexts. For example, the application of SB approaches, both from a top-down and bottom-up perspective, has the potential to be impactful in the context of prostate cancer health disparities.

## Figures and Tables

**Figure 1 nutrients-14-05197-f001:**
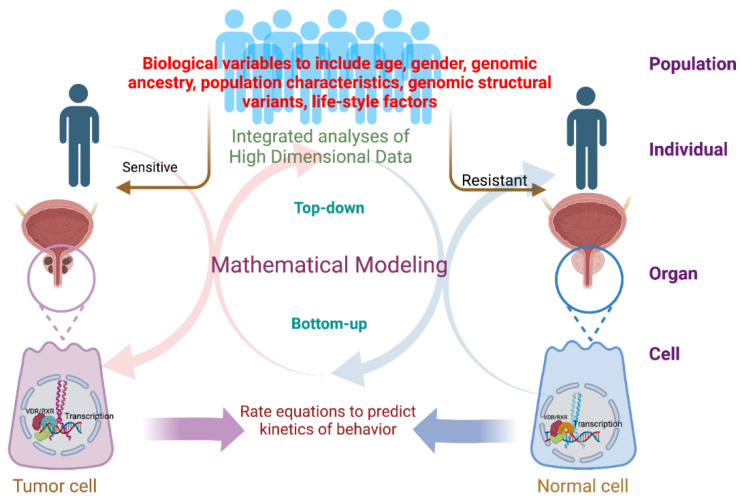
**A workflow for top-down and bottoms-up system biology (SB) approaches to address prostate cancer health disparities.** For top-down SB there is a plethora of high dimensional data including germline structural variants, population characteristics as serum-borne, and other clinical data that can be integrated through a range of approaches to deliver insights into how the prostate could either be either sensitive or resistant to the growth restraint properties of the VDR. From the bottoms-up perspective, more focused kinetic models can be developed for how the VDR functions and is disrupted between normal and tumor cells. Combining these approaches has the potential to develop a holistic understanding of how the VDR functions in the prostate gland and how this is impacted by genomic ancestry.

## Data Availability

Not applicable.

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
