# Peer review of "Vitamin D and Systems Biology"

_nutrients, 2022, doi:10.3390/nu14245197_

Round 1

Reviewer 1 Report

I found this review to be quite nice and comprehensive in regards to an overview of systems biology and what it means for the vitamin D field. My only comment would be for the authors to include a schematic figure or two to break up the dense text, but otherwise the review is good. 

Author Response

Reviewer #1

I found this review to be quite nice and comprehensive in regards to an overview of systems biology and what it means for the vitamin D field. My only comment would be for the authors to include a schematic figure or two to break up the dense text, but otherwise the review is good. 

Reply.

We thank the reviewer for supplying such positive comments and we have now generated Figure 1, which we believe encapsulates the essence of the review, and we hope the reviewer sees the figure as satisfactory and helpful to the reader.

Reviewer 2 Report

The review Hussain, Yates, and Campbell provide a well written account of systems biology and inquiry into how such strategies have been used to examine vitamin D biology. Although several examples were brought forward the impact of this review to move the field forward could be strengthened.  Here are some considerations:

1) It was not clear that systems biology approaches expanded our understanding of the role of vitamin D in prostate cancer as described in section 2.2-1. Certainly correlative studies that vitamin D affects systems that could impacts prostate cancer are supportive - but it was not clear if any specific studies have used systems modeling to make this case. 

2) Perhaps the review could be strengthened if the authors identified and specified knowledge gaps that could be addressed with systems biology approaches.

3) From a systems approach, perhaps it could be looked into if also adding in discussion on the unique biology involving VDR ligand hydroxylation states and affects on VDR system biology.

4) in the opinion of this reviewer, the second paragraph takes a very strong viewpoint on the limitations of reductionist science. Again just an opinion, but a softer take might be better appreciated by some readers, particularly if they share the viewpoint that the two approaches need not be antagonistic, but rather, complimentary.

Author Response

Reviewer #2

The review Hussain, Yates, and Campbell provide a well written account of systems biology and inquiry into how such strategies have been used to examine vitamin D biology. Although several examples were brought forward the impact of this review to move the field forward could be strengthened.  Here are some considerations:

1) It was not clear that systems biology approaches expanded our understanding of the role of vitamin D in prostate cancer as described in section 2.2-

Reply.

We have now updated the text starting at the bottom on P.12 (in red) and going through to the Summary (P.14) to stress how combined bottom-up and top-down modelling, which has not been undertaken for the VDR in prostate cancer, could reveal significant insight into VDR function in prostate cancer cells and itss contribution to prostate cancer health disparities.

2). Certainly correlative studies that vitamin D affects systems that could impacts prostate cancer are supportive - but it was not clear if any specific studies have used systems modeling to make this case. 

Reply.

The reviewer is correct that there certainly haven’t been any bottom-up modeling of VDR function in any system to date. We have now edited section 3.1 to emphasize the potential at the same time acknowledging that this hasn’t been undertaken. This potential is also illustrated in the new figure (Figure 1)

3) Perhaps the review could be strengthened if the authors identified and specified knowledge gaps that could be addressed with systems biology approaches.

Reply

Perhaps we’re not fully understanding what the reviewer is highlighting, as in a general sense this would be a massive topic. However, we feel the updated section 3.1 and Figure 1 somewhat address this by illustrating to the reader what would be gained.

4) From a systems approach, perhaps it could be looked into if also adding in discussion on the unique biology involving VDR ligand hydroxylation states and affects on VDR system biology.

Reply

This is a great suggestion from the reviewer, and it’s an obvious oversight on out behalf for which we apologize. We’ve addressed this in part by adding sections at the top of p.7, and a line in the first paragraph on p.11. Given the review is already long and the topic of hydorxlases is also covered in other chapters in this single topic issue, we feel this is sufficient to guide the reader.

5) in the opinion of this reviewer, the second paragraph takes a very strong viewpoint on the limitations of reductionist science. Again just an opinion, but a softer take might be better appreciated by some readers, particularly if they share the viewpoint that the two approaches need not be antagonistic, but rather, complimentary.

Reply

We thank the reviewer from helping us to be less confrontational and more inclusive! We’ve re-written the paragraph to reflect a more balanced view.

Reviewer 3 Report

The authors described systems biology concepts and how it can be applied to studying the vitamin D receptor functions. However, I did not identify important research aspects. What was the guiding question? what is the purpose of the review? What is the methodology used to extract articles? To be considered for publication in the scientific paper format, I suggest that authors frame a question that can be answered by the systems biology approach and perform a systematic review. 

Author Response

Reviewer 3

1) The authors described systems biology concepts and how it can be applied to studying the vitamin D receptor functions. However, I did not identify important research aspects.

What was the guiding question and purpose of the review?

Reply

The text was aimed at being a review of systems biology approaches, and in that regard somewhat of an introduction to an audience who are familiar with the biology of the VDR, but les familiar with systems approaches. Our goal was to illustrate how might systems biology approaches be applied to understand VDR function and dysfunction in disease. We have now more clearly stated this in the abstract and section 2 is retitled. Also, section 3.1 has been re-written, partly in line with the comments of Reviewer 2, to illustrate how systems biology approaches could be combined to address challenges in understanding the health disparities in prostate cancer experienced by African American men.

2) What is the methodology used to extract articles?

Reply

In p.10 we refer to an imbalance of where research is undertaken by disease site and where from an unbiased perspective there is the most compelling evidence of where VDR signaling maybe disrupted. Our paper reference #59 details the search terms and summary tables. We felt it was inappropriate to redo as it’s only a couple of years later and the work would be repetitious. That said, the same search terms were undertaken as detailed in that reference and the answer remained the same.  no correct to redo (although checked and #s are very similar) p.10

3) To be considered for publication in the scientific paper format, I suggest that authors frame a question that can be answered by the systems biology approach and perform a systematic review. 

Reply

We have now stressed, in the abstract, that the purpose of this review is to illustrate what systems biology approaches represent (for an audience who will most likely not be systems biologists) and what the review aims to achieve

Round 2

Reviewer 2 Report

The authors made sufficient responses to this reviewer's concerns. However, the article cannot be accepted until references are placed in the final paragraphs.

Author Response

Many apologies! these are now done. It was an oversight

Reviewer 3 Report

The authors answered the questions satisfactorily and made appropriate changes in the text, which made it sustainable for publication in its current form.

Author Response

Thank you to the reviewer. We're grateful for the revision suggestions which have clearly improved the review